# TRIM56-mediated monoubiquitination of cGAS for cytosolic DNA sensing

Gil Ju Seo[1], Charlotte Kim[1], Woo-Jin Shin [1], Ella H. Sklan[1,2], Hyungjin Eoh[1] & Jae U. Jung [1]

Intracellular nucleic acid sensors often undergo sophisticated modifications that are critical for the regulation of antimicrobial responses. Upon recognition of DNA, the cytosolic sensor cyclic GMP-AMP (cGAMP) synthase (cGAS) produces the second messenger cGAMP, which subsequently initiates downstream signaling to induce interferon-αβ (IFNαβ) production. Here we report that TRIM56 E3 ligase-induced monoubiquitination of cGAS is important for cytosolic DNA sensing and IFNαβ production to induce anti-DNA viral immunity. TRIM56 induces the Lys335 monoubiquitination of cGAS, resulting in a marked increase of its dimerization, DNA-binding activity, and cGAMP production. Consequently, TRIM56-deficient cells are defective in cGAS-mediated IFNαβ production upon herpes simplex virus-1 (HSV-1) infection. Furthermore, TRIM56-deficient mice show impaired IFNαβ production and high susceptibility to lethal HSV-1 infection but not to influenza A virus infection. This adds TRIM56 as a crucial component of the cytosolic DNA sensing pathway that induces anti-DNA viral innate immunity.

[1] Department of Molecular Microbiology and Immunology, Keck School of Medicine, University of Southern California, Los Angeles, CA 90033, USA. [2] Department of Clinical Microbiology and Immunology, Sackler School of Medicine, Tel-Aviv University, Tel-Aviv, 69978, Israel. Correspondence and requests for materials should be addressed to J.U.J. (email: jaeujung@med.usc.edu)

Pathogens are detected by host pattern recognition receptors (PRRs) that sense various microbial motifs collectively known as pathogen-associated molecular patterns (PAMPs) and subsequently elicit antimicrobial innate immune responses[1,2]. Microbe-derived nucleic acids are potent PAMPs that elicit PRR-mediated host immune responses[3]. The recognition of invading RNA viruses by cytoplasmic sensors (RIG-I and MDA5) and endosomal toll-like receptors (TLRs) has been extensively characterized[3]. The appearance of naked DNA in the cytoplasm of mammalian cells also triggers DNA sensor-mediated signal transduction[4]. Many cytosolic DNA sensors have been identified, including AIM2[5–7], DAI[8], DDX41[9], DNA-PK[10], IFI16[11], a form of RNA polymerase III[12,13], and cyclic GMP-AMP (cGAMP) synthase (cGAS)[14,15]. Particularly, recent studies have shown that cGAS functions as the primary cytosolic DNA sensor that triggers production of type I interferons (IFNs) and other inflammatory cytokines, such as tumor necrosis factor-α and interleukin-6, upon DNA transfection and DNA virus infection[14,15]. Following activation, cGAS converts ATP and GTP into the dinucleotide cGAMP[16–18]. cGAMP is a second messenger that binds to stimulator of interferon genes (STING), which in turn induces the recruitment of TANK-binding kinase 1 (TBK1) and interferon regulatory factor-3 (IRF-3)[18,19]. Then, the TBK-mediated activation of the IRF-3 pathway induces the expression of type I IFNs. Thus cGAS-mediated DNA sensing signals through various adaptor molecules to ultimately induce potent antiviral innate immunity.

Since both self and non-self DNA can activate intracellular DNA sensors, this DNA sensing pathway must be tightly regulated to prevent harmful activity arising from unrestrained signaling[20]. Given the central role of the cGAS pathway in the innate immune response to viral infections, it is expected that various modulations and modifications to cGAS control its activity. We have previously reported that the autophagy protein Beclin-1 negatively regulates cGAS activity[21]. Beclin-1 directly interacts with cGAS and suppresses cGAMP synthesis and signaling. On the other hand, this interaction enhances the autophagy-mediated degradation of cytosolic pathogen DNA to avoid persistent immune stimulation. Several posttranslational modifications, including phosphorylation and glutamylation, have been reported to play critical roles in regulating the cGAS-STING pathway[22]. Glutamylation of cGAS impairs its DNA binding and enzymatic activity[23]. We have also shown that Akt kinase suppresses cGAS enzymatic activity by phosphorylating its carboxyl-terminal enzymatic domain. This suppresses the subsequent antiviral cytokine production and leads to increased DNA virus replication[24]. These modulations fine-tune the IFN-mediated antiviral pathway to ultimately ensure that the host-DNA-sensing innate immune response is kept in balance after responding to stimuli, such as DNA virus infections.

Protein ubiquitination controls a large number of cellular processes, including protein degradation, DNA repair, chromatin remodeling, cell-cycle regulation, endocytosis, kinase signal pathways, and others[25]. The interaction between endoplasmic reticulum ubiquitin ligase RNF185 and cGAS specifically catalyzes the K27-linked polyubiquitination of cGAS, which promotes its enzymatic activity[26]. Members of the tripartite motif (TRIM) E3 ubiquitin ligase family have arisen as key molecules in antiviral immunity, either as direct restriction factors of viral replication or as regulators of nucleic acid sensing pathways[27]. TRIM25 and TRIM4 mediate the K63-linked ubiquitination that activates RIG-I cytosolic RNA sensor[28]. On the other hand, TRIM14 inhibits the degradation of cGAS DNA sensor mediated by selective autophagy receptor p62, which promotes innate immune responses. TRIM56 has been shown to be a restriction factor of several RNA viruses (influenza virus, yellow fever virus, dengue virus, and

bovine viral diarrhea virus) both in an E3 ligase-dependent and -independent manner[29–31]. Furthermore, *Salmonella typhimurium* SopA HECT-type E3 ligase targets TRIM56 and TRIM65 to stimulate RIG-I and MDA5 innate immune receptors, which subsequently modulates inflammatory responses[32]. However, since previous studies were limited to in vitro cell culture assays, the in vivo role of TRIM56 still remains unclear. While an initial report described a direct role of TRIM56 in the STING-mediated double-stranded DNA sensing pathway, a later study convincingly disputed that TRIM56 has no role in the STING-mediated pathway, suggesting that alternative mechanisms or functions of TRIM56 should be explored[33,34].

In order to determine the specific in vivo role of TRIM56, we generated TRIM56-deficient cells and mice and identify that TRIM56 directly targets cGAS, rather than STING or its downstream signaling molecules, to confer DNA sensing-mediated innate immune responses. TRIM56 interacts with the amino-terminal regulatory domain of cGAS and this interaction promotes the Lys335 monoubiquitination of cGAS, resulting in the increase of its cGAMP production. Consequently, TRIM56 deficiency leads to a dramatic reduction of cGAS-mediated IFNαβ production upon herpes simplex virus type 1 (HSV-1) infection but not upon influenza A virus (IAV) infection. Our in vitro and in vivo study indicates that the TRIM56-induced mono-ubiquitination of cGAS is important for efficient cytosolic DNA sensing in response to DNA virus infection.

## Results

**TRIM56 interacts with cGAS.** Mass spectrometry of a purified FLAG-tagged cGAS complex identified the IFN-inducible E3 ubiquitin ligase TRIM56 with an apparent molecular mass of 81 kDa as a cGAS-binding partner (Fig. 1a). Co-immunoprecipitation showed the specific interaction of cGAS-FLAG with either exogenous or endogenous TRIM56 in HEK293T cells (Fig. 1b, c). This interaction was specific since cGAS-FLAG did not interact with TRIM25 under the same conditions (Fig. 1b). The single-molecule pull-down (SiMPull) technique combines the principles of a conventional pull-down assay with single-molecule fluorescence microscopy and enables direct visualization of individual protein–protein interactions[35]. Specifically, this approach immobilizes protein complexes from lysed cells directly on a coverslip, which is then studied under a total internal reflection fluorescence microscope. This SiMPull assay also revealed that TRIM56 effectively bound to cGAS (Fig. 1d). cGAS contains an amino (N)-terminal regulatory domain (RD, a.a. 1–160), a central nucleotidyl transferase (NTase) domain (a.a. 161–330), and a carboxyl (C)-terminal domain (CTD, a.a. 331–522), whereas TRIM56 contains an N-terminal RING E3 ligase domain (a.a. 1–96), B-box domain (a.a. 96–205), a coiled-coil domain (a.a. 223–307), and the C-terminal NHL homologous region (a.a. 307–755)[21,29]. We further examined whether TRIM56 directly interacted with cGAS by using bacterially purified cGAS and TRIM56 purified from HEK293T cells. Full-length (FL) cGAS fused with maltose-binding protein (MBP) interacted with TRIM56. However, both MBP-cGAS C-terminal region (a.a. 161–522) and MBP alone failed to interact with TRIM56 (Supplementary Fig. 1a). Mapping study showed that the N-terminal RD of cGAS and the C-terminal NCL1-HT2A-Lin41 (NHL) homologous region of TRIM56 were responsible for their interaction (Fig. 1e). Collectively, these results showed the specific interaction between cGAS and TRIM56.

**TRIM56 is essential for cGAS-mediated DNA sensing activity.** TRIM56 has been previously shown to directly modulate STING

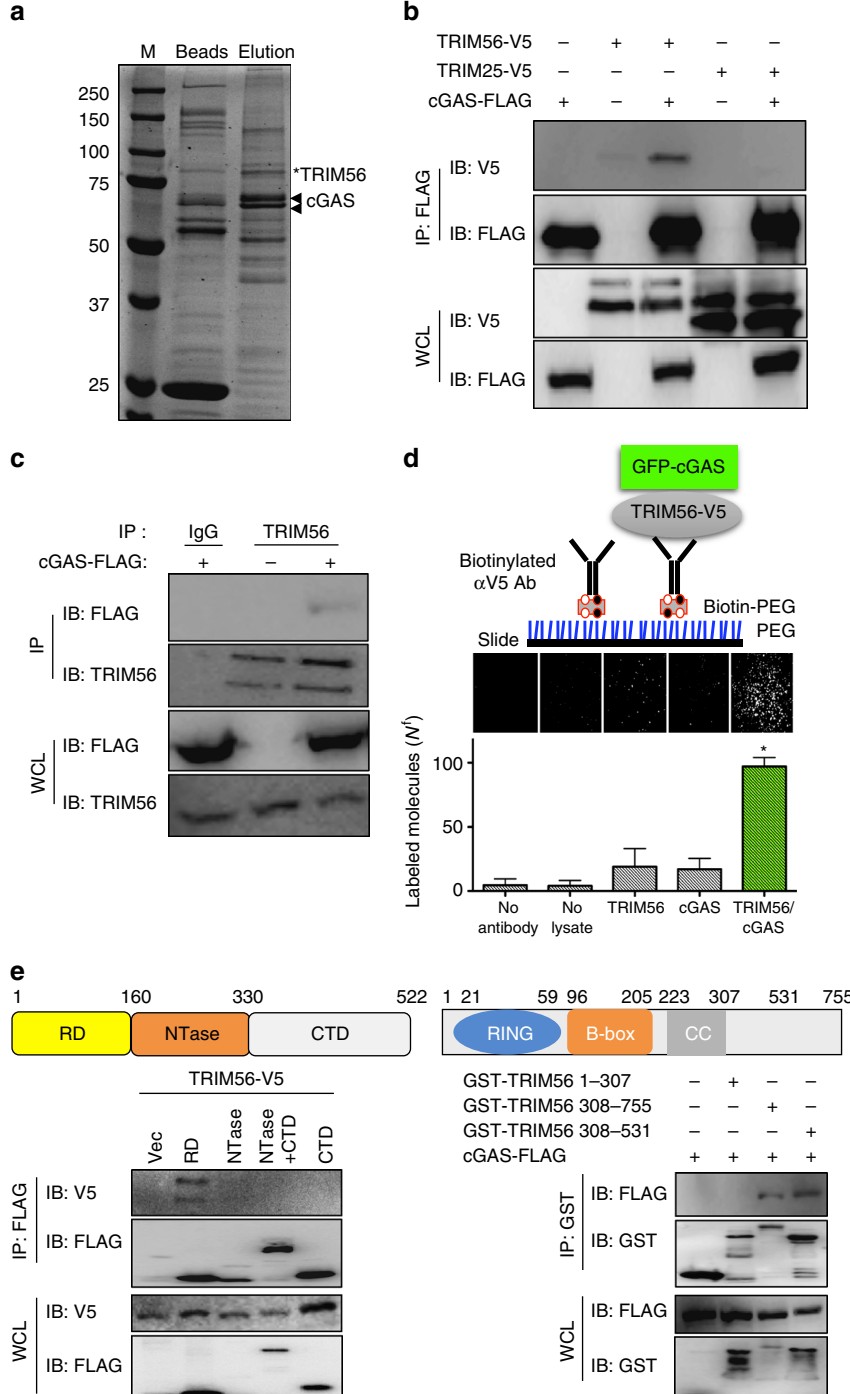

**Fig. 1** TRIM56 interacts with cGAS. **a** HEK293T cells were transiently transfected with cGAS-FLAG and protein complexes were purified from cell lysates with anti-FLAG M2 beads. The immunoprecipitates were eluted with FLAG peptides and fractionated using SDS-PAGE followed by Coomassie blue staining. The bands for TRIM56 and cGAS identified by mass spectrometry are indicated on the right. **b** Co-immunoprecipitation analysis of the interaction between cGAS-FLAG and TRIM56-V5. HEK293T cells expressing cGAS-FLAG were co-transfected with TRIM56-V5 or TRIM25-V5. cGAS-FLAG immunoprecipitates were probed with FLAG and V5 antibody and whole-cell lysates (WCLs) were probed with FLAG and V5 antibody, as indicated. **c** cGAS interacts with endogenously expressed TRIM56. HEK293T cells were transfected with cGAS-FLAG. Rabbit IgG or TRIM56 immunoprecipitates were probed with FLAG antibody, and WCLs were probed with TRIM56 and FLAG, as indicated. **d** cGAS-FLAG was transiently coexpressed with TRIM56-V5 in HEK293T cells. TRIM56-V5 was pulled down for SiMPull analysis. Top panel: schematic depiction of cGAS SiMPull; middle panel: representative GFP fluorescence images; bottom panel: average number of molecules per imaging area ($N_f$). **e** The N-terminal regulatory domain of cGAS interacts with the C-terminal NHL homologous region of TRIM56. Left panel: the regulatory domain of cGAS binds to TRIM56. WCLs of HEK293T cells transfected with TRIM56-V5 and cGAS truncation mutants were used for immunoprecipitation and immunoblotting with the indicated antibodies. Right panel: the C-terminus domain (308–531) of TRIM56 interacts with cGAS. WCLs of HEK293T cells transfected with cGAS-FLAG and TRIM56 truncation mutants were used for immunoprecipitation and immunoblotting with the indicated antibodies. Data are representative of two independent experiments **a–e**. Error bars indicate mean ± s.d. of n = 3. *P < 0.05 versus control using Student's t-test (**d**). Full blots are shown in Supplementary Fig. 10

activity to induce IFN production in response to double-stranded DNA stimulation[33]. To test the role of TRIM56 in STING-mediated DNA sensing, human monocyte THP-1 cells were stably transfected with two different TRIM56-specific short-hairpin RNAs (shRNAs) to deplete TRIM56 expression (Fig. 2a). *IFNβ* mRNA levels were measured upon herring testis (HT) DNA transfection or cGAMP stimulation. Depletion of *TRIM56* expression led to the significant reduction of HT-DNA mediated *IFNβ* mRNA expression but did not affect cGAMP-mediated *IFNβ* mRNA expression (Fig. 2b). These results suggest that, unlike the previous report[33], TRIM56 plays an important function in the cytosolic DNA sensing pathway upstream of STING. In order to further investigate the role of TRIM56 in cGAS-mediated DNA sensing, we used the CRISPR-Cas9 technology to disrupt the *TRIM56* gene in L929 mouse fibroblasts *cGAS(−)* cells[36] and complemented with mouse cGAS-FLAG (Fig. 2c). Interestingly, *TRIM56(+) cGAS(−)*, *TRIM56(−) cGAS(+)*, and *TRIM56(−) cGAS(−)* cells failed to induce *IFNβ* mRNA expression upon HT-DNA transfection, whereas these cells effectively induced *IFNβ* mRNA expression upon cGAMP stimulation (Fig. 2d). In contrast, wild-type *TRIM56(+) cGAS(+)* L929 cells were capable of inducing *IFNβ* mRNA expression upon either HT-DNA or cGAMP stimulation (Fig. 2d). Furthermore, *TRIM56(+) cGAS(−)*, *TRIM56(−) cGAS(+)*, and *TRIM56(−) cGAS(−)* cells failed to induce *IP-10* mRNA expression upon HSV-1ΔICP34.5 infection, whereas wild-type (WT) L929 cells were capable of inducing *IP-10* mRNA expression upon HSV-1ΔICP34.5 infection (Supplementary Fig. 1c). Specifically, the HSV-1ΔICP34.5 mutant strain was used since the GADD34 homology-containing ICP34.5 effectively counteracts the type I IFN response by binding four cellular proteins, Beclin-1, TBK1, protein phosphatase 1α, and eukaryotic translation initiation factor 2α[37–39].

To further dissect the role of TRIM56, WT, *TRIM56(+) cGAS(−)*, *TRIM56(−) cGAS(+)*, or *TRIM56(−) cGAS(−)* L929 cells were stably transfected with an ISRE-IFNβ promoter-driven secreted luciferase reporter, followed by stimulation with HT-DNA, pdAdT, pdGdC, pI:C, or cGAMP. *TRIM56(+) cGAS(−)*, *TRIM56(−) cGAS(+)*, or *TRIM56(−) cGAS(−)* cells also failed to induce ISRE-IFNβ promoter-luciferase expression upon HT-DNA, pdA:dT, or pdG:dC stimulation, but they effectively induced ISRE-luciferase expression upon pI:C or cGAMP stimulation (Fig. 2e). Finally, these cells were infected with HSV-1 or an RNA virus, Sendai virus (SeV), followed by measurement of the ISRE-IFNβ promoter-luciferase activity. While wild-type L929 cells markedly induced ISRE-IFNβ promoter-luciferase expression upon HSV-1ΔICP34.5 infection, *TRIM56(+) cGAS(−)*, *TRIM56(−) cGAS(+)*, or *TRIM56(−) cGAS(−)* L929 cells showed defective ISRE-IFNβ promoter-luciferase expression under the same conditions (Fig. 2f). In contrast, all L929 cell lines effectively induced ISRE promoter-luciferase expression upon SeV infection (Fig. 2f). After mock or HT-DNA treatment, WT, *TRIM56(+) cGAS(−)*, *TRIM56(−) cGAS(+)*, or *TRIM56(−) cGAS(−)* cell lysates were added into digitonin-permeabilized THP1-Lucia™ ISG reporter cells to perform a cGAMP bio-assay (Fig. 2g). Stimulation with WT L929 cell lysates led to a significant increase of cGAMP-mediated IFN responsiveness of THP1-Lucia™ ISG reporter cells (Fig. 2h). In contrast, stimulation with *TRIM56(+) cGAS(−)*, *TRIM56(−) cGAS(+)*, or *TRIM56(−) cGAS(−)* cell lysates showed little or no increase of cGAMP-mediated IFN responsiveness (Fig. 2h).

To comprehensively understand the role of TRIM56 on the cGAS-STING pathway upon DNA or RNA virus infection, we observed the co-localization of exogenously expressed cGAS or STING with endogenous TRIM56. Both endogenous TRIM56 and exogenous green fluorescent protein (GFP)-cGAS effectively

co-localized at the indicated foci upon HT-DNA stimulation or HSV-1 infection but not upon SeV infection. However, STING did not co-localize with either cGAS or TRIM56 under any conditions (Supplementary Fig. 2). Collectively, the results indicate that TRIM56 is essential for cGAS-mediated DNA sensing activity.

**TRIM56 E3 ligase induces the monoubiquitination of cGAS.** It was intriguing to detect that, upon treatment of proteasome inhibitor MG132 and deubiquitinase inhibitor n-ethylmaleimide, a small portion of cGAS underwent a migration shift with an apparent molecular weight of ~8 kDa in HEK293T cells (Fig. 3a). This migration shift was more obvious upon TRIM56 over-expression, suggesting that TRIM56 E3 ubiquitin ligase may induce the monoubiquitination of cGAS (Fig. 3a). To examine the level of cGAS monoubiquitination with or without TRIM56 expression, the densities of the higher molecular-weight cGAS bands were quantitated in three independent experiments. This showed that cGAS monoubiquitination was increased ~6 fold upon TRIM56 expression (Supplementary Fig. 3a, b). To test the potential TRIM56-mediated monoubiquitination of cGAS, HEK293T cells were transfected with cGAS-FLAG with or without TRIM56-V5, treated with MG132 and n-ethylmaleimide, and subjected to immunoprecipitation with anti-FLAG antibody, followed by immunoblotting with a monoubiquitin-specific VU-1 antibody. A portion of cGAS was readily detected by VU-1 antibody and this VU-1 antibody reactivity was further increased by TRIM56 expression (Fig. 3b). These monoubiquitinated forms of cGAS migrated near 75 kDa (Supplementary Fig. 3c). To further confirm cGAS monoubiquitination, we used the previously described L929 cell lines. cGAS-FLAG was stably expressed in *TRIM56(+) cGAS(−)* or *TRIM56(−) cGAS(−)* L929 cells, and purified cGAS-FLAG protein was then subjected to immuno-blotting with an anti-VU-1 antibody. This experiment showed that monoubiquitination of cGAS-FLAG could be only detected in *TRIM56*-expressing cells (Fig. 3c and Supplementary Fig. 4a). Finally, an in vitro ubiquitination assay using purified cGAS demonstrated that TRIM56 E3 ligase effectively induced the monoubiquitination of cGAS favorably with UbcH5a or UbcH5c as an E2 enzyme (Fig. 3d and Supplementary Fig. 4b). Taken together, these results collectively show that TRIM56 E3 ligase induces the monoubiquitination of cGAS.

**cGAS K335 monoubiquitination is important for DNA sensing.** To identify the specific lysine residue(s) of cGAS that are targeted by TRIM56 for monoubiquitination, cGAS-FLAG was co-expressed with TRIM56 in HEK293T cells and partially purified for multi-dimensional liquid chromatography coupled with tandem mass spectrometry (Supplementary Fig. 5a). Three potential ubiquitination sites were identified: Lys278, Lys335, and Lys350 (Supplementary Fig. 5b). To test their role in cGAS monoubiquitination, each of these lysine residues was mutated to arginine (Lys278R, Lys335R, and Lys350R). These mutants were co-expressed with TRIM56 and blotted with anti-VU-1 antibody. While the K278R or K350R mutants underwent TRIM56-mediated monoubiquitination as efficiently as cGAS WT, the cGAS K335R mutant completely lost its TRIM56-mediated monoubiquitination (Fig. 4a). This indicates that the K335 residue of cGAS is the primary site for TRIM56-induced monoubiquitination.

To test the role of TRIM56-induced K335 monoubiquitination of cGAS in IFN signaling, cGAS WT, K278R, K335R, or K350R were co-transfected with TRIM56 and the IFNβ promoter-luciferase reporter. The cGAS K335R mutant showed weak IFNβ promoter activation, and its activity was not increased by

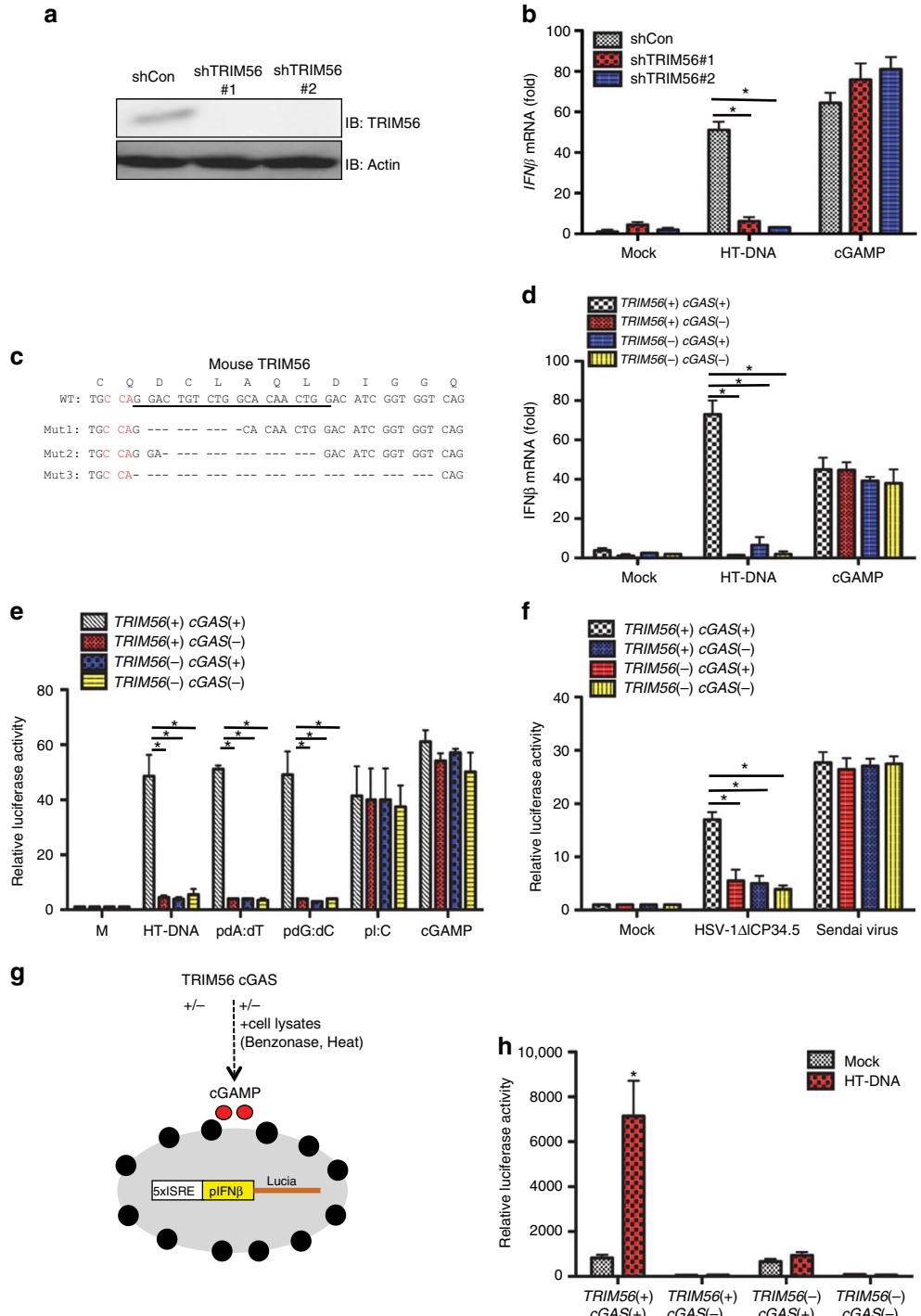

**Fig. 2** TRIM56 is required for cGAS-dependent signal transduction and acts upstream of STING. **a** THP-1 cells were stably transfected with control shRNA, TRIM56#1 shRNA, or TRIM56#2 shRNA. Cell lysates were detected with the indicated antibodies. **b** THP-1 cells used in **a** were stimulated with HT-DNA (2 μg/ml) and cGAMP (3 μg/ml). The expression of IFNβ mRNA was measured using real-time PCR. **c** Generation of TRIM56$^{-/-}$ L929 cells. CRISPR-Cas9 targeting: the CRISPR target site in exon 1 of TRIM56 is underlined, and the protospacer adjacent motif (PAM) is shown in red. Deletions in the three TRIM56 alleles, each of which results in a frameshift, are shown by the black dashes. **d** cGAS-deleted L929 cells with or without TRIM56 deletion were complemented with wild-type cGAS-FLAG and stimulated with HT-DNA (2 μg/ml) and cGAMP (3 μg/ml). The expression of *IFNβ* mRNA was measured using real-time PCR. **e** L929 cell lines used in **d** were stably transfected with ISREIFNβ-Lucia reporter genes. Cells were stimulated with HT-DNA (2 μg/ml), poly dA:dT (1 μg/ml), poly dG:dC (1 μg/ml) polyI:C (1 μg/ml), and cGAMP (3 μg/ml). Luciferase activity was measured 18 h after stimulation. **f** L929 cell lines used in **e** were infected with HSV-1ΔICP34.5 (MOI = 5) and Sendai virus. Luciferase activity was measured 18 h after stimulation. **g** Schematic representation of the cGAMP bio-assay. Relative cGAMP activity was measured using THP1-Lucia ISG cells that generate luciferase in response to cGAMP. **h** cGAMP bio-assay. L929 cell lines used in **d** were stimulated with HT-DNA (2 μg/ml) for 9 h. Extracts of the cells were prepared, DNase- and heat-treated, and incubated with permeabilized THP-1-Lucia ISG cells for 18 h. Relative luciferase activity was measured. Data in **a**, **h** are representative of two independent experiments. Data in **b**, **d**–**f** are representative of three independent experiments. Error bars in **b**, **d**–**f**, **h** indicate mean ± s.d. of n = 3 (**b**, **d**–**e**) and n = 2 (**f**, **h**). *P < 0.05, versus control using Student's t-test (**b**, **d**–**f**, **h**). Full blots are shown in Supplementary Fig. 10

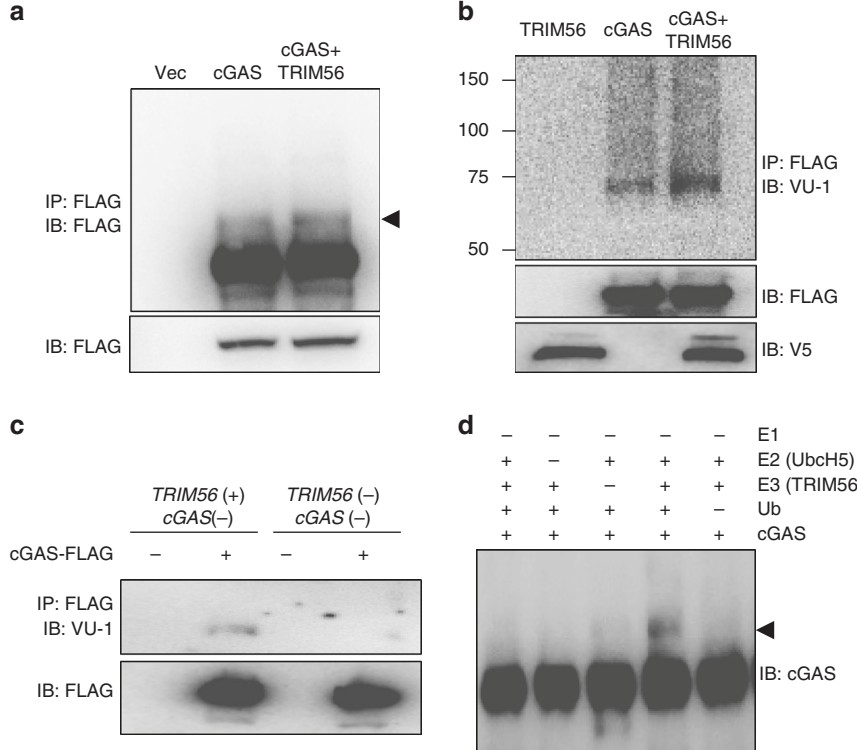

**Fig. 3** TRIM56 mono-ubiquitinates cGAS. **a**, **b** HEK293T cells were transfected with the indicated plasmids. Twenty-four hours after transfection, whole-cell lysates (WCLs) were used for immunoprecipitation and immunoblotting, as indicated. **c** WCLs from L929 cell lines were used for immunoprecipitation and immunoblotting, as indicated. **d** Immunoblot analysis of the in vitro ubiquitination with the indicated combinations of ubiquitin, purified cGAS, E1, UbcH5 E2 enzymes, and TRIM56 (E3). Data in **a**, **b** are representative of three independent experiments. Data in **c**, **d** are representative of two independent experiments. Full blots are shown in Supplementary Fig. 10

TRIM56 expression (Fig. 4b). By striking contrast, cGAS WT, K278R, or K350R showed strong IFNβ promoter activation, which was further enhanced by TRIM56 expression (Fig. 4b). To further delineate the role of cGAS K335 monoubiquitination, cGAS(−) L929 cells were stably complemented with cGAS WT or the K335R mutant followed by HT-DNA stimulation or HSV-1ΔICP34.5 infection. cGAS WT expression in cGAS(−) L929 cells robustly induced IFNβ mRNA production upon HSV-1ΔICP34.5 infection or HT-DNA stimulation, whereas expression of the cGAS K335R mutant led to a minimal induction of IFNβ mRNA under the same conditions (Fig. 4c). To test whether the K335R mutation affected cGAS NTase activity, bacterially purified cGAS WT or K335R mutant was subjected to an in vitro cGAMP production assay. This showed that the cGAS K335R mutant produced a comparable level of cGAMP to cGAS WT in vitro (Fig. 4d), suggesting that the lack of DNA sensing activity of cGAS K335R mutant in cells is due to the loss of TRIM56-induced monoubiquitination, not the loss of NTase activity. These results demonstrate that TRIM56 catalyzes the mono-ubiquitination of cGAS at the lysine 335 residue, which is critical for its DNA sensing activity.

**TRIM56 enhances cGAS dimerization and DNA-binding activity**. cGAS forms an oligomeric complex with bound DNA and subsequently undergoes switch-like conformational changes at the activation loop[40,41]. Localization of the K335 residue on the interaction surface suggested that the TRIM56-induced mono-ubiquitination might facilitate cGAS dimerization and/or DNA-binding activity. To investigate whether TRIM56 affected the cGAS dimerization or oligomerization, we performed SiMPull

photo bleaching analysis. Besides the protein–protein interaction shown in Fig. 1d, the SiMPull technique also provides a measure of protein stoichiometry directly determined from the stepwise pattern of fluorophore photo bleaching, as well as protein interactions with other compounds, such as nucleic acids, small molecule ligands, and lipids[42]. For the fluorophore photo bleaching assay, cGAS-GFP fusion was expressed in HEK293T cells with or without TRIM56, followed by photo bleaching and counting the fluorescence spots. This showed that a majority of WT cGAS and K335R mutant was present as a monomer without TRIM56 expression (Fig. 4e and Supplementary Fig. 6). Approximately 20% of cGAS WT underwent dimerization upon TRIM56 expression, whereas the cGAS K335R mutant showed little or no increase in dimerization under these conditions (Fig. 4e). cGAS dimerization increases DNA-binding activity and cGAS bound to DNA undergoes a conformational change to catalyze cGAMP synthesis[40,41]. Thus we tested whether TRIM56 increases cGAS DNA-binding activity. Rhodamine-labeled DNA was co-transfected into vector-, cGAS-, or TRIM56/cGAS-expressing cells. Further SiMPull analysis indicated that overexpression of TRIM56 efficiently increased the double-stranded DNA (dsDNA)-binding activity of cGAS WT but not the cGAS K335R mutant (Fig. 4f). These results suggest that TRIM56 expression induces cGAS oligomerization, which subsequently increases its cGAS DNA-binding activity.

**TRIM56 has a critical role in in vivo HSV-1 infection**. To investigate the in vivo antiviral activity of TRIM56, we generated TRIM56[−/−] mice in which the first exon of TRIM56 was spliced and fused with a LacZ cassette, thus abrogating TRIM56

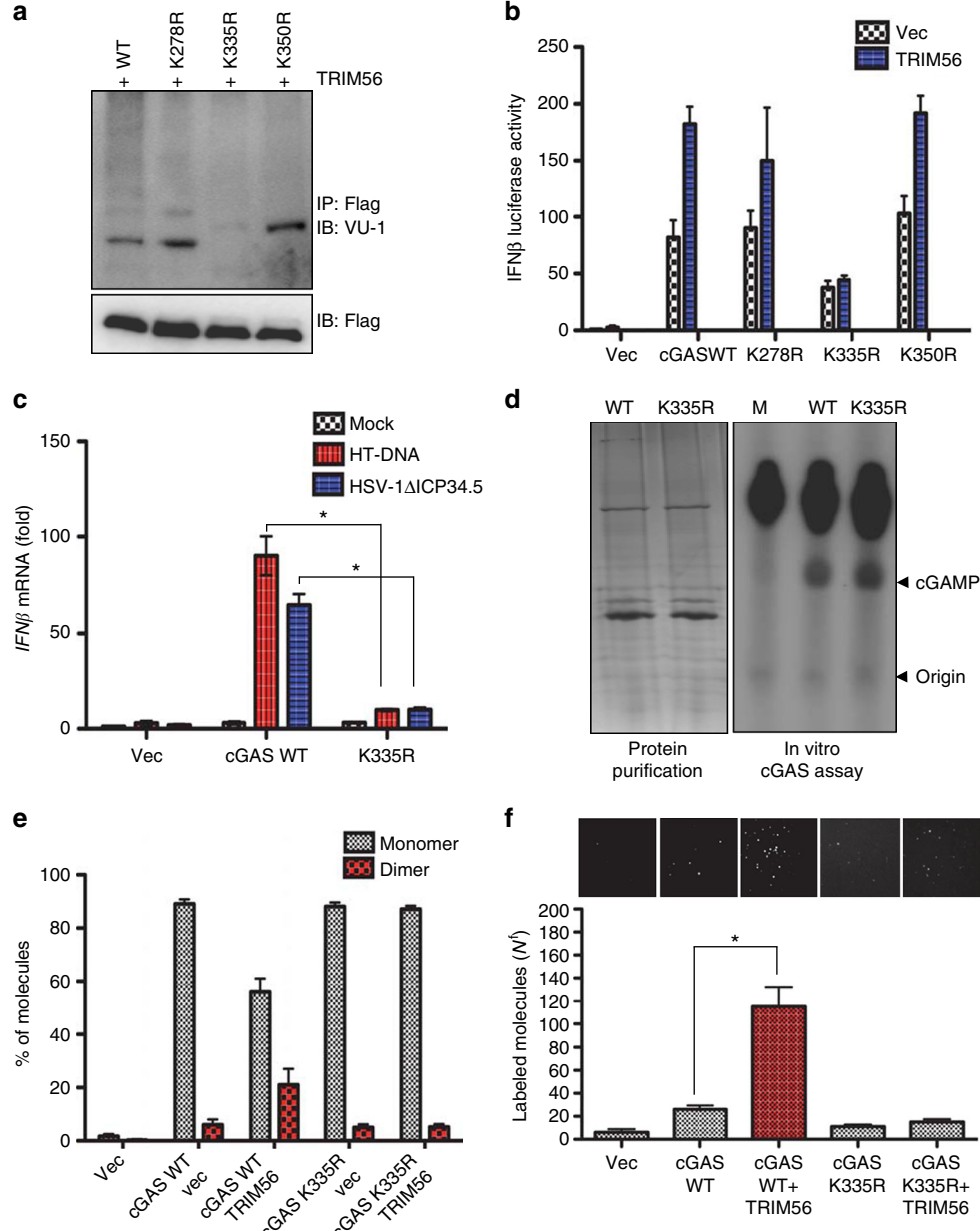

**Fig. 4** TRIM56 increases cGAS dimerization and DNA-binding ability. **a** HEK293T cells were transfected with cGAS WT or cGAS K→R mutants. WCLs were used for immunoprecipitation and immunoblotting, as indicated. **b** IFNβ promoter activity in 293T cells, which were co-transfected with TRIM56 and cGAS WT or K→R mutants. **c** L929 cGAS$^{-/-}$ cells were stably complemented with empty vector, mouse cGAS WT, or cGAS K335R. Cells were stimulated with HT-DNA or infected with HSV-1ΔICP34.5 (MOI = 5). *IFNβ* mRNA was analyzed by RT-PCR. **d** In vitro enzymatic assays were performed in the presence of P$^{32}$-α-GTP with amino acids 141–507 of mouse cGAS purified from engineered *E. coli* BL21 strain. The left panel shows the purified cGAS WT and cGAS K335R proteins. cGAMP production was analyzed by TLC and autoradiography. The bottom arrow shows the spotted origin and the top arrow shows the migrated cGAMP. **e** The relative ratio of observed bleaching steps for monomeric cGAS-GFP and dimeric cGAS-GFP pulled down from cell lysates. HEK293T cells were transfected with vector, cGAS WT, cGAS WT/TRIM56, cGASK335R, or cGAS335R/TRIM56, and the ratio was calculated by counting the photobleaching steps from a GFP-cGAS pull-down. For determining the stoichiometry, traces were manually scored for the number of bleaching steps. More than 300 traces were scored to reliably identify the photobleaching step distribution. See Supplementary Fig. 6. **f** Rhodamine-labeled double-stranded DNA bound by cGAS on the PEG-coated surface. Top panel: representative fluorescent DNA images bound by cGAS pulled down from cell lysates from vector; cGAS, cGAS- and TRIM56-expressing cells; cGASK335R-; or cGASK335R- and TRIM56-expressing cells. Rhodamine-labeled pdA: dT (1 μg/ml) was co-transfected into vector, cGAS, TRIM56/cGAS-expressing cells, cGASK335R, or cGAS335R/TRIM56-expressing cells, respectively. Bottom panel: average number of molecules per imaging area ($N_f$). Error bars indicate mean ± s.d. *$P < 0.05$, Student's *t*-test. Data in **a**, **d** are representative of two independent experiments. Data in **b**, **c**, **e**, **f** are representative of three independent experiments. Error bars **b**, **c**, **e**, **f** indicate mean ± s.d. of $n = 3$. *$P < 0.05$, versus control using Student's *t*-test (**c**, **f**). Full blots are shown in Supplementary Fig. 10

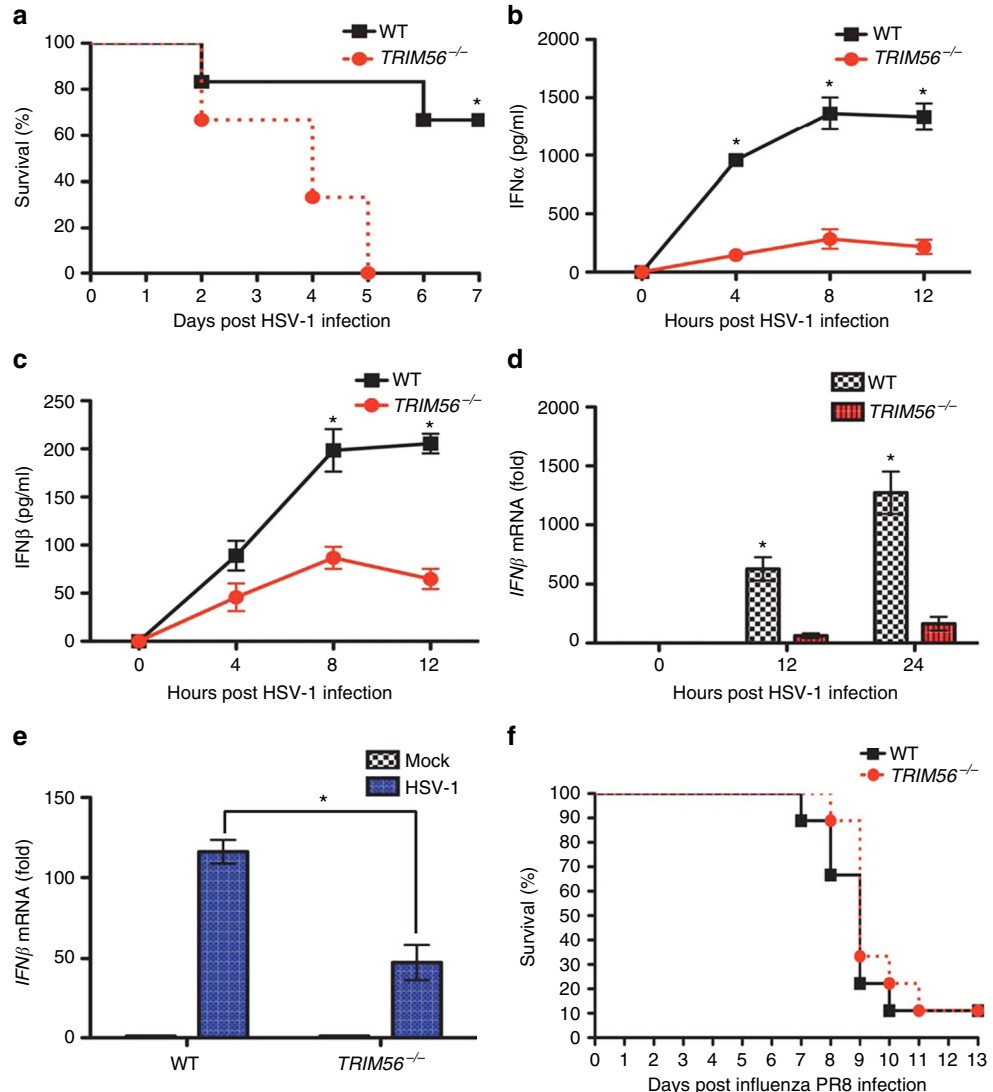

**Fig. 5** $TRIM56^{-/-}$ mice are susceptible to HSV-1 infection but not to IAV infection. **a** HSV-1 ($1.8 \times 10^8$ pfu/mouse) was injected peritoneally to WT or $TRIM56^{-/-}$ mice ($n = 6$ each). Survival rates were monitored for 7 days. **b**, **c** Sera were collected from WT or $TRIM56^{-/-}$ mice injected with HSV-1 at the indicated time points, and IFNα or IFNβ concentration was measured by ELISA ($n = 3$ each at each time point). Total peritoneal cavity cells (**d**) and peritoneal macrophages (**e**) were isolated from WT or $TRIM56^{-/-}$ HSV-1-infected mice ($n = 3$ for each time point). $IFN\beta$ mRNA levels in these cells were measured using RT-PCR. **f** Influenza PR8 (1000 pfu/mouse) was administered intranasally to WT or $TRIM56^{-/-}$ mice ($n = 9$ each). Survival rates were monitored for 13 days. Error bars indicate mean $\pm$ s.d. *$P < 0.05$, Student's $t$-test. Data in **a–f** are representative of two independent experiments. Error bars in **b–e** indicate mean $\pm$ s.d. of $n = 3$. *$P < 0.05$, versus control using Student's $t$-test (**a–e**)

expression (Supplementary Fig. 7). While $TRIM56^{-/-}$ mice displayed lower fertility compared to WT mice, they showed no obvious phenotypes. WT or $TRIM56^{-/-}$ mice were infected with HSV-1 with $1.8 \times 10^8$ plaque-forming units/mouse and their survival was monitored. We found that $TRIM56^{-/-}$ mice ($n = 6$) rapidly lost weight, showed severe disease symptoms, and died over a period of 5 days (Fig. 5a). In contrast, 67% of WT mice ($n = 4$) showed minor symptoms, recovered their lost weight, and survived; 33% of them ($n = 2$) showed moderate symptoms and died over a period of 6–7 days postinfection (Fig. 5a). Moreover, $TRIM56^{-/-}$ mice had considerably lower blood IFNα and IFNβ levels in blood than WT mice following HSV-1 infection (Fig. 5b, c). Peritoneal cavity cells of HSV-1-infected $TRIM56^{-/-}$ mice showed much lower $IFN\beta$ mRNA levels than those of HSV-1-infected WT mice (Fig. 5d). Finally, peritoneal macrophages were isolated from WT or $TRIM56^{-/-}$ mice infected with HSV-1 and tested for $IFN\beta$ mRNA levels. Peritoneal macrophages from

$TRIM56^{-/-}$ mice had significantly lower $IFN\beta$ mRNA levels upon HSV-1 infection compared to those from WT mice ($p < 0.05$, Fig. 5e). In contrast to the previous in vitro study[29], $TRIM56^{-/-}$ mice ($n = 9$) showed nearly no difference from WT mice ($n = 9$) in response to in vivo IAV PR8 infection; both mice rapidly lost weight and died over a period of 11 days (Fig. 5f and Supplementary Fig. 8a). These results demonstrate the critical role of TRIM56 in the in vivo IFN response to HSV-1 infection but not to IAV infection.

**TRIM56 has a critical role in in vitro HSV-1 infection**. To further assess the critical role of TRIM56 in the cGAS-mediated DNA sensing pathway, WT or $TRIM56^{-/-}$ mouse bone marrow-derived macrophages (BMDMs) were infected with HSV-1 or HSV-1ΔICP34.5 (Fig. 6a). While WT BMDMs robustly induced $IFN\beta$ mRNA expression upon HSV-1 or HSV-1ΔICP34.5

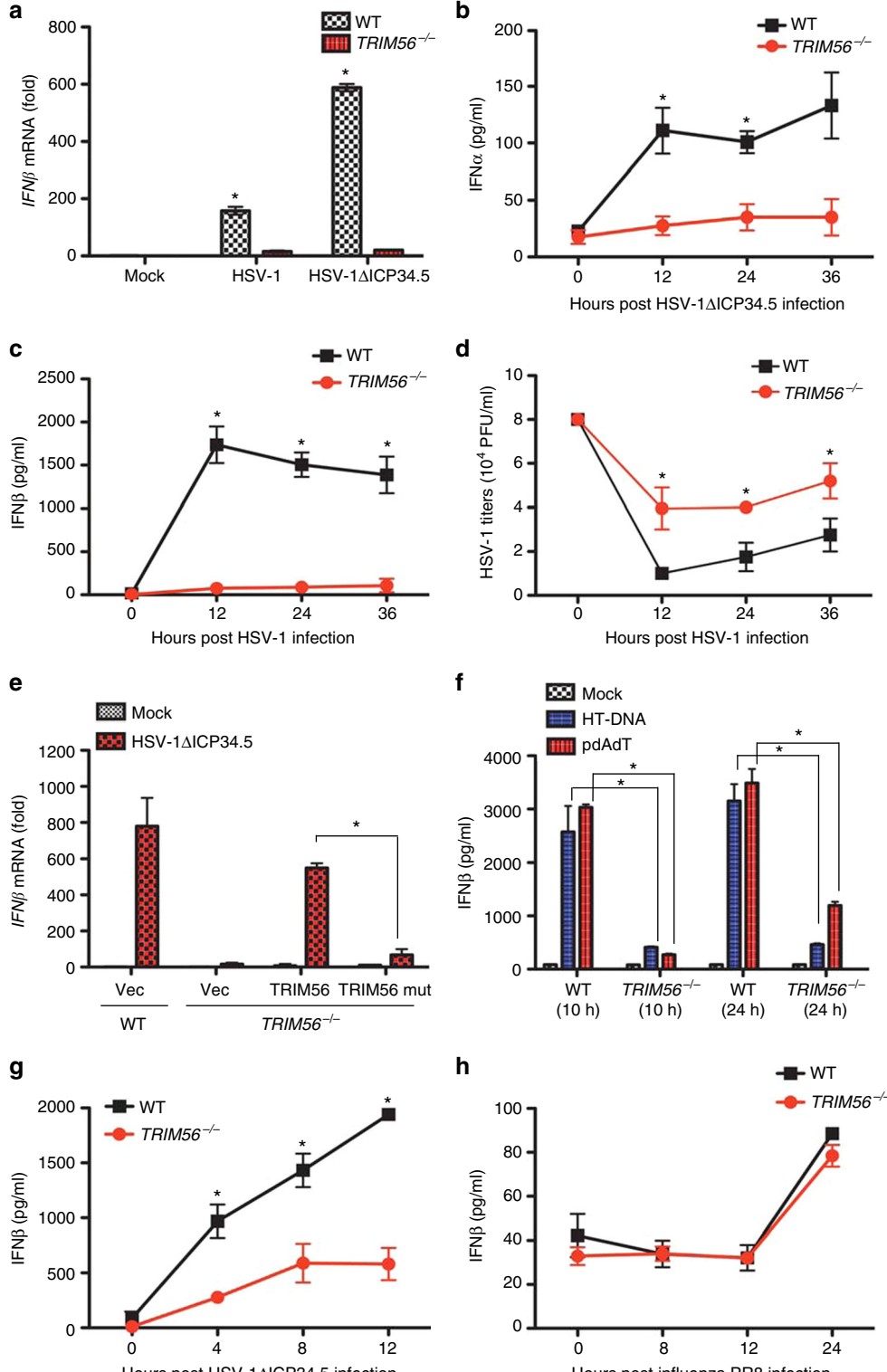

**Fig. 6** TRIM56 is required for IFN induction in response to HSV-1 infection but not to IAV infection in primary cells. **a** RT-PCR analysis of *IFNβ* mRNA in BMDMs from WT and *TRIM56*[-/-] mice at 18 h after mock, HSV-1, or HSV-1ΔICP34.5 infection. ELISA of IFNα (**b**) or IFNβ (**c**) in BMDMs from WT and *TRIM56*[-/-] mice at the indicated time points after HSV-1ΔICP34.5 or HSV-1 infection. **d** WT or *TRIM56*[-/-] BMDMs were infected with HSV-1 (MOI = 0.1). Viral supernatants were collected at the indicated time points, and virus titers were determined using a plaque assay on Vero cells. **e** RT-PCR analysis of *IFNβ* mRNA in WT or *TRIM56*[-/-] BMDMs infected with lentivirus-derived vector, TRIM56 WT, or TRIM56 Mut. Results are presented relative to empty vector-expressing and mock-infected WT cells. **f** ELISA of IFNβ in MEFs from WT and *TRIM56*[-/-] embryos. MEFs were stimulated with HT- DNA (2 μg/ml) or polydA:dT (1 μg/ml). **g** ELISA of IFNβ in BMDCs from WT and *TRIM56*[-/-] mice at the indicated time points following infection with HSV-1ΔICP34.5. **h** ELISA of IFNβ in BMDMs from WT and *TRIM56*[-/-] mice at the indicated time points following infection with IAV PR8. Data in **a**–**g** are representative of three independent experiments. Data in **h** are representative of two independent experiments. Error bars in **a**–**h** indicate mean ± s.d. of *n* = 3. \**P* < 0.05 versus control using Student's *t*-test (**a**–**h**)

infection, *TRIM56⁻/⁻* BMDMs displayed no increase of *IFNβ* mRNA expression under the same conditions (Fig. 6a). Time-course experiments also showed that *TRIM56⁻/⁻* BMDMs induced little or no IFNα/β production upon HSV-1ΔICP34.5 infection, whereas WT BMDMs robustly induced IFNα/β production (Fig. 6b, c). Consistently, significant higher titers of HSV-1 were measured in *TRIM56⁻/⁻* BMDMs compared to WT BMDMs (Fig. 6d). To determine whether the TRIM56 E3 ligase enzymatic activity is required to induce *IFNβ* mRNA expression upon HSV-1ΔICP34.5 infection, we generated the enzymatically dead mutant (Mut) of TRIM56 by replacing the cysteine 21 and 24 residues with serines and complemented *TRIM56⁻/⁻* BMDMs with lentivirus-derived *TRIM56*WT or *TRIM56 Mut* (Supplementary Fig. 8b, Fig. 6e). These results showed that, while expression of *TRIM56*WT in *TRIM56⁻/⁻* BMDMs efficiently induced *IFNβ* mRNA expression, expression of *TRIM56 Mut* led to a minimal increase of *IFNβ* mRNA expression (Fig. 6e). We tested IFN responses in WT and *TRIM56⁻/⁻* mouse embryonic fibroblast (MEFs) and bone marrow-derived dendritic cells (BMDCs). Similarly, we observed that *TRIM56⁻/⁻* MEFs failed to induce IFNβ production following HT-DNA or pdA:dT stimulation compared to WT MEFs (Fig. 6f). In addition, time-course experiments showed that *TRIM56⁻/⁻* BMDCs weakly induced IFNβ production compared to WT BMDCs upon HSV-1ΔICP34.5 infection (Fig. 6g). In agreement with previous in vivo infection results (Fig. 5f), *TRIM56⁻/⁻* BMDMs also showed nearly no difference from WT BMDMs in response to IAV PR8 infection; both BMDMs robustly produced IFNβ (Fig. 6h).

We further examined whether TRIM56 was involved in the host's response to positive- and negative-strand RNA viruses. WT and *TRIM56⁻/⁻* BMDMs showed no significant differences in the magnitude of IFNβ mRNA induction upon infection of positive-strand RNA flavivirus such as Zika virus and Dengue virus (Supplementary Fig. 9a). Furthermore, a positive-strand RNA alphavirus (Sindbis virus) and a negative-strand RNA virus (Parainfluenza virus) also induced IFNβ mRNA at similar levels between WT and *TRIM56⁻/⁻* BMDMs (Supplementary Fig. 9b). We also showed that WT and *TRIM56⁻/⁻* BMDMs showed similar levels of IFNα production upon infection with Sindbis virus or Parainfluenza virus (Supplementary Fig. 9c). Furthermore, WT and *TRIM56⁻/⁻* BMDMs showed similar levels of IFNβ production upon infection of Flavivirus (Zika virus or Dengue virus), Sindbis virus, or Parainfluenza virus (Supplementary Fig. 9d, e). WT and *TRIM56⁻/⁻* BMDMs were infected with Sindbis virus and viral titers were then measured at different time points using a standard plaque-formation assay on Vero cells. This showed similar viral titers of Sindbis virus in WT and *TRIM56⁻/⁻* BMDMs (Supplementary Fig. 9f). Finally, WT and *TRIM56⁻/⁻* BMDMs also showed similar viral RNA loads of ZIKV MR766 strain or H/PF2013 strain during 48 h infection periods (Supplementary Fig. 9g). These comprehensive results demonstrate that TRIM56 is critical for the IFN response against DNA virus infection but appears to be not required for negative- and positive-strand RNA virus infection.

## Discussion

Much about the signal transduction triggered by cytosolic DNA has been studied over the past decades, leading to the discovery of a number of intracellular DNA sensors, including cGAS, DAI, IFI16, DDX41, and MRE11. Among these, cGAS has been recognized as a primary cytosolic DNA sensor that initiates STING-dependent downstream signaling in multiple cell types upon DNA stimulation. Given the demanding interest in cGAS DNA sensor activity, its regulatory

mechanisms have been also extensively investigated[22]. Particularly, posttranslational modifications, such as phosphorylation, ubiquitination, and sumoylation, have been shown to be central in regulating the cGAS-STING-mediated innate immune response[22]. Here we showed that TRIM56 induced the Lys335 monoubiquitination of cGAS, which resulted in a marked increase of its dimerization, DNA-binding activity, and cGAMP production of cGAS. Consequently, TRIM56-deficient cells and mice were defective in cGAS-mediated IFNαβ production upon HSV-1 infection but not to IAV infection. This indicates that TRIM56-induced monoubiquitination of cGAS is critical for cytosolic DNA sensing and IFNαβ production to induce anti-DNA viral immunity.

Several pieces of evidence substantiate the crucial function of TRIM56 in activating the cytosolic cGAS DNA sensing pathways. TRIM56 interacted with cGAS through its uncharacterized C-terminal NHL homology domain. The NHL repeat, named after NCL1-HT2A-Lin41, is found in a large number of serine/threonine protein kinases in a diverse range of pathogenic bacteria. Interestingly, similar to TRIM56, the *Drosophila melanogaster* Brat protein also carries a N-terminal RING domain, one or two B-box motifs, a coiled-coil region, and a C-terminal NHL domain[29]. This C-terminal β-propeller-shaped NHL domain has been shown to bind RNA in a sequence-specific manner to direct differentiation of neuronal stem cells[43,44]. It is possible that TRIM56 also uses the C-terminal NHL homologous region to tether the cGAS for ubiquitination. On the other hand, cGAS utilizes its N-terminal RD for interaction with TRIM56. While the physiological function of the N-terminal RD of cGAS is largely uncharacterized, a recent report has demonstrated that the N-terminal RD helps full-length cGAS to expand the binding range on DNA and increases dsDNA-binding efficiency, enzyme activity, and activation of STING/IRF3-mediated cytosolic DNA signaling[45]. Consistent to this, TRIM56 interaction resulted in the marked increase of cGAS dimerization, DNA-binding activity, and cGAMP production, indicating that TRIM56 is an important signaling molecule for the DNA sensing pathway.

cGAS forms an oligomeric complex with bound DNA and subsequently undergoes switch-like conformational changes at the activation loop[40,41]. We identified cGAS as a substrate of TRIM56 and also defined TRIM56 as a potent enhancer of cGAS signaling activity. TRIM56 induced the monoubiquitination of the cGAS Lys335 residue located on the interaction surface, and this monoubiquitination resulted in the marked increase of cGAS dimerization and DNA-binding activity. We primarily observed the monoubiquitinated form of cGAS from in vitro and in vivo TRIM56-mediated cGAS ubiquitination assays. Zhang et al.[40] have shown that the Lys335 residue of mouse cGAS not only mediates DNA binding but also interacts with the Glu398 of the adjacent protomer. Thus we speculate that the TRIM56-induced monoubiquitination of cGAS Lys335 may reinforce the interaction between two cGAS protomers, leading to stable dimerization and subsequently enhancing DNA-binding ability. Future analysis of the cGAS–TRIM56 complex will provide new structural understanding of how TRIM56-mediated monoubiquitination affects cGAS dimerization and DNA-binding activity.

While TRIM56 was reported to have a direct role in the STING-mediated dsDNA sensing pathway, this was later convincingly disputed[33,34]. Here we also demonstrated that depleting *TRIM56* expression significantly reduced *IFNβ* mRNA expression induced by DNA stimulation or HSV-1 infection, but not *IFNβ* mRNA expression induced by cGAMP stimulation (Fig. 2). This indicates that TRIM56 is a genuine antiviral modulator that functions upstream of STING, ultimately protecting the hosts

from microbial or DNA viral infection. Next, TRIM56 has been shown to be a restriction factor of several RNA viruses, including IAV, in both an E3 ligase-dependent manner and -independent manner[29–31]. However, all previous studies were limited to in vitro cell line assays. Using TRIM56$^{-/-}$ primary cells and mice, we unambiguously demonstrate that TRIM56 is critical for the in vitro and in vivo IFN response against HSV-1 infection but not against IAV infection. However, we cannot rule out that, besides targeting cGAS, TRIM56 may have additional functions that direct responses to certain RNA virus infections. In fact, TRIM56 has been also shown to interact with TRIF to enhance the TLR3 signaling pathway[46] as well as S. typhimurium SopA HECT-type E3 ligase to modulate inflammatory responses[32]. Kane et al.[47] has recently reported an ISG screening that uncovered the TRIM56 gene as an important player in STING-independent antiviral activity. Furthermore, TRIM56 is deleted in some splenic marginal zone lymphomas[48], and its expression is considerably altered in primary effusion lymphomas[49]. Further work will help identify the cGAS-dependent or -independent functions of TRIM56 in host antimicrobial immunity and cancer.

In summary, we demonstrate that TRIM56 is a critical positive factor that targets the cGAS-mediated DNA sensing pathway. Although hosts have developed numerous strategies to recognize and block viral infection, viruses continue to overcome these obstacles. Thus it is not surprising that certain viral factors inhibit critical regulators of the host nucleic acid sensing pathway, such as TRIM56. Additional study of TRIM56 will provide new insights into this everlasting arms race between viruses and their hosts.

## Methods

**Mice**. WT C57BL/6J were purchased from the Jackson Laboratory. TRIM56 KO mice (Wellcome Trust Sanger Institute, UK) were housed in specific pathogen-free barrier facilities. Male and female mice aged 6–14 weeks were used. Primary bone marrow and peritoneal cavity cells were collected from WT and TRIM56 knockout mice for BMDMs and BMDCs. All experiments were approved and done according to the guidelines of the Institutional Animal Care and Use Committee at the University of Southern California. Sample sizes were estimated based on previous similar studies for innate immunity. Animal experiments were not randomized or blinded.

**Reagents, constructs and cell culture**. The reagents were purchased from the following companies: HT-DNA and anti-FLAGM2-agarose (Sigma); pdAdT, pdGdC, pI:C (Invivogen); and cGAMP (Biolog, Germany). To clone mouse TRIM56, we purchased an expressing plasmid (Addgene). Mouse TRIM56-V5 constructs were created by subcloning the PCR products into pEF-IRES-puro. Mutagenesis by overlap extension PCR was used to create mouse TRIM56 Mut, which was subcloned into pEF-IRES-puro. Primers used for cloning are presented in Supplementary Table 1. The constructs were sequenced by Genewiz and were identical to reference TRIM56 sequences except for the defined mutations. HEK293T (ATCC; CRL-3216), L929 WT (ATCC; CCL-1), L929 cGAS knockout[36], and Vero cells (ATCC; CCL-81) were obtained from American Type Culture Collection (ATCC) and cultured in Dulbecco's modified Eagle's medium (DMEM; Gibco-BRL) containing 4 mM glutamine and 10% fetal bovine serum (FBS). HEK293T, L929, and Vero cells are authenticated by ATCC and were not further validated in our laboratory. Transient transfections were performed with Lipofectamine 2000 (Invitrogen) according to the manufacturer's instructions. Cell lines stably expressing cGAS were generated using a standard selection protocol with 2 µg/ml of puromycin (Invitrogen). Target sequences to generate stable cell lines are presented in Supplementary Table 2. All cell lines used during these studies tested negative in bi-monthly mycoplasma screening.

**Mass spectrometry**. HEK293T cells were collected 48 h after transfection with cGAS-FLAG and/or TRIM56-V5 and lysed with NP-40 buffer (50 mM HEPES, pH 7.4, 150 mM NaCl, 1 mM EDTA, 1% [vol/vol] NP-40) supplemented with complete protease inhibitor cocktail (Roche). Post centrifugation, supernatants were mixed with a 50% slurry of FLAG M2 beads (Sigma, M8823), and the binding reaction mix was incubated for 4 h at 4 °C. Precipitates were washed extensively with lysis buffer. Proteins bound to FLAG M2 beads were eluted and separated in a NuPAGE 4–12% Bis-Tris gradient gel (Invitrogen). After Coomassie brilliant blue staining, protein bands corresponding to the cGAS-FLAG were excised and separately analyzed by ion-trap mass spectrometry at the Harvard Taplin Biological Mass Spectrometry Facility in Boston, MA[50].

**SiMPull assay**. Polyethylene glycol (PEG)-coated quartz slides with flow chambers were obtained according to previous protocols. A PEG surface was coated with Neutravidin (0.05 mg/ml) followed by anti-GFP or V5 (rabbit, 5 µg/ml) antibody conjugated with biotin (Rockland) and incubated for another 5 min in T50 (10 mM Tris pH 8 and 50 mM NaCl). Enhanced GFP-fused cGAS cell lysates were added to the antibody-coated surface and incubated for 5–10 min. The exposure time was 30 ms and the single-molecule signals were processed with a custom-edited IDL and Matlab program.

**Dual luciferase assay**. HEK293T cells were seeded into 12-well plates. Twenty-four hours later, the cells were transfected with 100 ng of IFNβ luciferase reporter plasmid and 20 ng of TK-renilla luciferase. In addition, 100 ng of plasmid encoding human or mouse cGAS, 50 ng of plasmid encoding STING, or 400 ng TRIM56 was transfected. Forty-eight hours after transfection, whole-cell lysates were prepared and subjected to the Dual-Glo luciferase assay according to the manufacturer's instructions (Promega). Results are presented with Renilla luciferase levels normalized by the firefly luciferase levels.

**Immunoblot and immunoprecipitation**. Cell lysates were collected in 1% NP-40 buffer and quantified by BCA protein assay (Thermo Scientific). Proteins were separated by sodium dodecyl sulfate (SDS)-polyacrylamide gel electrophoresis and transferred to polyvinylidene difluoride membrane (Bio-Rad) by semi-dry transfer at 25 V for 40 min. All membranes were blocked in 5% milk in Tris-Buffered Saline (TBST) and probed overnight with the indicated antibodies, which were diluted in the blocking buffer solution at 4 °C. Primary antibodies included mouse FLAG (Sigma, F3165), rabbit V5 (Bethyl Laboratories, A190), and mouse VU-1 (Life-Sensors, VU101). Horseradish peroxidase-conjugated secondary antibodies were incubated on membranes in 5% milk in TBST, and bands were developed with ECL reagent (Thermo Scientific) and imaged using a Fuji LAS-4000 imager.

For immunoprecipitation, cells were harvested and then lysed in 1% NP-40 buffer supplemented with a complete protease inhibitor cocktail (Roche). After preclearing with protein A/G agarose beads for 1 h at 4 °C, whole-cell lysates were used for immunoprecipitation with the indicated antibodies. A 50% slurry of FLAGM2 beads (Sigma, M8823) was added to 1 ml of cell lysates and incubated at 4 °C for 4 h. Immunoprecipitates were extensively washed five times with lysis buffer and eluted with SDS loading buffer by boiling for 5 min. Full-length uncropped blots are presented in Supplementary Fig. 10.

**Protein purification**. MBP-cGAS (mouse) fusion protein (151–522 AA) and full length were expressed in BL21 (DE3, RIPL strain). Escherichia coli were grown at 37 °C until reaching OD$_{600}$ = 0.6. The temperature was then shifted to 18 °C and grown overnight by adding 1 mM IPTG. Cells were collected by centrifugation and lysed with 1% NP40 buffer. Clarified lysates were mixed with Ni-NTA agarose (Invitrogen) and washed five times with lysis buffer prior to elution of protein using 150 mM imidazole. The concentrated protein was aliquoted and stored at −80 °C for the in vitro cGAS enzyme assay and binding assay.

**In vitro ubiquitination assay**. The in vitro ubiquitination assays using TRIM56 (E3) and MBP-cGAS as substrates were performed using a non-radioactive assay (Enzo Life Sciences). Briefly, assay components (10× Ubiquitinylation buffer, 1PP (100 U/ml), dithiothreitol (DTT) (50 mM), Mg$^2$+ATP, EDTA (50 mM), 20 × E1, 10 × E2, TRIM56 E3, cGAS, 20xBt-Ub) were added to 1.5 ml Eppendorf tubes and the tube contents were mixed gently. The mixture was incubated at 37 °C for 60 min. The assay was quenched by addition of 50 µl 2× non-reducing gel loading buffer. It proceeded directly to western blot.

**In vitro cGAS activity assay**. For in vitro cGAS enzymatic reactions, 1 µM MBP-cGAS or MBP-cGAS variant was mixed with 250 µM GTP and 10 µCi P$^{32}$ alpha phosphate radiolabed ATP in buffer [20 mM Tris-Cl (pH 7.5), 150 mM NaCl, 5 mM MgCl$_2$, 1 mM DTT]. After a 1 h incubation at 37 °C, the reaction was treated with 5 units of alkaline phosphatase (Roche) for 30 min to stop the reaction and 1 µl of reaction solution was spotted onto TLC plates (HPTLC silica gel 60 F254, 20 × 10 cm, cat. # 1.50628.001) with the solvent 1:1.5 [v/v] 1 M (NH$_4$)$_2$SO$_4$ and 1.5 M KH$_2$PO$_4$. To visualize the reaction, the TLC plates were air-dried and imaged using a Fuji Phosphor Imager.

**RNA extraction and real-time PCR**. Total RNA was isolated from cells with TriReagent (Sigma). cDNA synthesis was performed using the iScript cDNA Synthesis Kit (Bio-Rad), and quantitative PCR reaction was monitored with SYBR Green Supermix (Bio-Rad). Primers used for real-time PCR are presented in Supplementary Table 3.

**Enzyme-linked immunosorbant assay (ELISA)**. L929 cells were treated with HT-DNA. The cell culture supernatant was collected and measured for IFNα or IFNβ by sandwich ELISA using a mouse IFNα or IFNβ ELISA Kit (PBL Biomedical Laboratories) according to the manufacturer's protocols.

**Viral plaque assays**. HSV-1 strain 17+ were grown in Vero cells propagated in DMEM supplemented with 10% FBS and antibiotics. BMDM cells were inoculated with virus in DMEM supplemented with 2% FBS for 1 h. After incubation for the indicated times (0, 12, 24, and 36 h), culture medium was harvested and the viral titer was determined. Briefly, culture medium from BMDM cells was serially diluted in DMEM supplemented with 10% FBS and used to infect 80% confluent monolayers of Vero cells growing in six-well plates. After infection, the monolayers were rinsed and overlaid with 7% methylcellulose. Plaques were counted after 3–5 days.

**Statistical analysis**. All data were analyzed using the Prism 5 software (GradphPad Software, Inc). All data were analyzed using a two-tailed Student's $t$-test with a minimum of $n = 3$. $p$-Values $< 0.05$ were considered significant.

**Data availability**. The data that support the findings of this study are available in this article and its Supplementary Information files or from the corresponding author upon request.

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

## Acknowledgements
This work was partly supported by CA200422, CA180779, DE023926, AI073099, AI116585, the Hastings Foundation, and the Fletcher Jones Foundation (to J.U.J.). We thank Dr. Philip Kranzusch for providing cGAS bacterial expression constructs. We thank all of Jung's laboratory members for their discussions.

## Author contributions
G.J.S. and J.U.J. designed the experiments. G.J.S., C.K., W.-J.S., and E.H.S. performed the research. All the authors analyzed the data. G.J.S., C.K., and J.U.J. wrote and edited the manuscript.

## Additional information

**Competing interests:** The authors declare no competing financial interests.

