## [Peer Review File · Nature Communications]

Reviewer #1 (Remarks to the Author):

The manuscript entitled "TRIM56-mediated monoubiquitination of cGAS for cytosolic DNA sensing" by Seo and collaborators explore the involvement of the E3 ligase TRIM56 on cGAS dependent DNA detection and antiviral function. The authors perform a systematic analysis of TRIM56 and cGAS interaction, identifying the domains involved. Also, they characterize the type of ubiquitination involved in the named mechanism and a clear anti HSV function in vitro and in vivo using TRIM56 KO mice.

The experiments presented by the authors are very clear and show persuasive evidences of the proposed mechanism.

It is important to confirm that the proposed TRIM56/cGAS interaction is direct. For this the authors need to perform IP with bacteria-produced purified proteins in order to demonstrate the proposed interaction

Strikingly, a publication by the Akira lab (which the authors reference in the manuscript) showed also an antiviral effect of TRIM56 but through a different mechanism that involves interaction with the adaptor STING and its ubiquitination, which they show to be crucial for its interaction with TBK1 and the induction of type I IFN response.

Even though, Seo et al. propose a TRIM56 effect on cGAS only. They only show one experiment using cGAMPs as STING stimulator, showing no effect. In these experiments (Fig 2) the authors do not see an effect on cGAMP detection by STING. In order to confirm this hypothesis, they need to explore with more detail the effect of TRIM56 on the DNA sensing versus RNA sensing pathways, since both STING and cGAS have shown antiviral properties for DNA and RNA viruses, in particular STING which can crosstalk to the RIG-I like and DNA sensing pathways.

Visualization of endogenous TRIM56 in mock, DNA transfection, DNA virus infection and RNA virus infection conditions would help to clarify this. Also co-localization of cGAS and STING will give more relevance to the experiment and the paper.

Regarding the role of TRIM56/cGAS on virus detection, the authors only use SeV and Influenza (negative ssRNA viruses) as control. It has been shown that cGAS and STING play a role not only for DNA viruses but also for positive sense RNA viruses, but not for negative sense RNA viruses (Charles Rice lab, Schoggins et al Nature 2013). The authors should add a positive sense RNA virus and reevaluate the role of TRIM56 on infection and antiviral response

I also suggest to repeat the TRIM56 and STING immunoprecipitation experiment shown by the Akira lab and add it as supplementary figure.

Reviewer #2 (Remarks to the Author):

Comments

TRIM proteins have a number of biological functions including immunity, cell proliferation and differentiation, signal transduction and autophagy. Recent evidence indicates several TRIM proteins serve as regulators in several signalling pathways in natural immunity. One of the TRIM proteins, TRIM56, has been reported to be a crucial regulator for viral infection and interferon signals.

In this paper, the authors showed that TRIM56 regulates the interferon signalling pathway. The authors found that TRIM56 interacts with cGAS and mono-ubiquitinates cGAS at the Lys335 and that TRIM56 exhibits dimerization of cGAS and DNA-binding activity and cGAMP production. These findings suggest that TRIM52 is a critical molecule for the cytosolic DNA sensing.

These insights are relevant in understanding the novel roles of TRIM52 in regulation of the innate immune system. This manuscript contains some important issues, such as the molecular insight of TRIM52 in the ubiquitination on cGAS. Although the experiments have been well performed, there may be some lacks of important controls and overstatements of results that preclude publication of the manuscript in the present form.

Specific points

1. Fig. 3a: The results is not clear. The authors should quantitate the monoubiquitinated forms and analyze statistically using three individual experiments.
2. Fig. 3b: To confirm that the band or smear is monoubiquitinated cGAS, the authors should show the molecular weight markers.
3. Fig. 3d: We can see the slight poly(or multi)-ubiquitinated cGAS at lane 4. Can the authors rule out that polyubiquitinated cGAS by TRIM56 is important in DNA sensing system?
4. In this study, the authors did not directly show that monoubiquitinated cGAS by TRIM56 enhances the dimerization and the binding activity to target DNA. It is very important to show the data substantiating this molecular insights.

Minor points:

1. The Method section (in vitro cGAS activity assay and others): carefully check again. The authors put "space" between numeric values and unit.

Reviewer #1 (Remarks to the Author):

The manuscript entitled “TRIM56-mediated monoubiquitination of cGAS for cytosolic DNA sensing” by Seo and collaborators explore the involvement of the E3 ligase TRIM56 on cGAS dependent DNA detection and antiviral function. The authors perform a systematic analysis of TRIM56 and cGAS interaction, identifying the domains involved. Also, they characterize the type of ubiquitination involved in the named mechanism and a clear anti HSV function in vitro and in vivo using TRIM56 KO mice. The experiments presented by the authors are very clear and show persuasive evidences of the proposed mechanism.

It is important to confirm that the proposed TRIM56/cGAS interaction is direct. For this the authors need to perform IP with bacteria-produced purified proteins in order to demonstrate the proposed interaction

→ According to the reviewer's suggestion, we further examined whether TRIM56 directly interacted with cGAS by using bacterially purified cGAS and TRIM56 purified from 293T cells. As shown in the figure below, full-length (FL) cGAS fused with Maltose-Binding Protein (MBP) interacted with TRIM56. However, both MBP-cGAS C-terminal region (a.a.161-522) and MBP alone failed to interact with TRIM56. We have included this result to the revised manuscript. White arrows indicate MBP (left), MBP-cGAS C-terminal 161-522aa (middle) and MBP-cGAS full-length (right), respectively.

Strikingly, a publication by the Akira lab (which the authors reference in the manuscript) showed also an antiviral effect of TRIM56 but through a different mechanism that involves interaction with the adaptor STING and its ubiquitination, which they show to be crucial for its interaction with TBK1 and the induction of type I IFN response. Even though, Seo et al. propose a TRIM56 effect on cGAS only. They only show one experiment using cGAMPs as STING stimulator, showing no effect. In these experiments (Fig 2) the authors do not see an effect on cGAMP detection by STING. In order to confirm this hypothesis, they need to explore with more detail the effect of TRIM56 on the DNA sensing versus RNA sensing pathways, since both STING and cGAS have shown antiviral properties for DNA and RNA viruses, in particular STING which can crosstalk to the RIG-I like and DNA sensing pathways. Visualization of endogenous TRIM56 in mock, DNA transfection, DNA virus infection and RNA virus infection conditions would help to clarify this. Also co-localization of cGAS and STING will give more relevance to the experiment and the paper.

→ To comprehensively understand the role of TRIM56 on the cGAS-STING pathway upon DNA or RNA virus infection, we observed the co-localization of exogenously expressed cGAS or

STING with endogenous TRIM56. Both endogenous TRIM56 and exogenous GFP-cGAS effectively co-localized at the indicated foci upon HT-DNA stimulation or Herpes Simplex Virus-1 (HSV-1) infection, but not upon Sendai virus infection (please see the figure with arrowheads below). However, STING did not co-localize with either cGAS or TRIM56 in any conditions. We have included this result in the revised manuscript.

Regarding the role of TRIM56/cGAS on virus detection, the authors only use SeV and Influenza (negative ssRNA viruses) as control. It has been shown that cGAS and STING play a role not only for DNA viruses but also for positive sense RNA viruses, but not for negative sense RNA viruses (Charles Rice lab, Schoggins et al Nature 2013). The authors should add a positive sense RNA virus and reevaluate the role of TRIM56 on infection and antiviral response

→ As the reviewer suggested, we examined whether TRIM56 is involved in the host's response to positive-sense RNA viruses. As shown in Figure (a) below, WT and TRIM56^{-/-} BMDMs showed no significant differences in the magnitude of IFN β mRNA induction upon infection of positive-strand RNA Flavivirus such as Zika virus or Dengue virus. Furthermore, another positive-strand RNA alphavirus (Sindbis virus) or a negative-strand RNA virus (Parainfluenza virus) also induced IFN β mRNA at similar levels between WT and TRIM56^{-/-} BMDMs (Figure b).

We also showed that WT and TRIM56^{-/-} BMDMs showed similar levels of IFN α production upon infection with Sindbis virus or Parainfluenza virus (Figure c). Furthermore, WT and TRIM56^{-/-} BMDMs showed similar levels of IFN β production upon infection with Flavivirus (Zika virus or Dengue virus), Sindbis virus or Parainfluenza virus (Figure d, e).

WT and TRIM56^{-/-} BMDMs were infected with Sindbis virus and viral titers were measured at different time points using a standard plaque formation assay on Vero cells. It

showed similar viral titers of Sindbis virus in WT and TRIM56^{-/-} BMDMs (Figure f). Finally, WT and TRIM56^{-/-} BMDMs also showed similar viral RNA loads of ZIKV MR766 strain and ZIKV H/PF2013 strain during 48h infection periods (Figure g). These results indicate that TRIM56 is not required for the host response to control positive-sense RNA viruses. We have included these results to our revised manuscript.

I also suggest to repeat the TRIM56 and STING immunoprecipitation experiment shown by the Akira lab and add it as supplementary figure.

→ As the reviewer suggested, we conducted a co-immunoprecipitation experiment to determine the interaction between cGAS or STING with TRIM56. TRIM56 co-immunoprecipitated with cGAS, not STING, even in stringent binding and washing conditions (500mM NaCl). This shows that cGAS specifically binds to TRIM56. We have included these results to the revised manuscript.

Reviewer #2 (Remarks to the Author):

Comments

TRIM proteins have a number of biological functions including immunity, cell proliferation and differentiation, signal transduction and autophagy. Recent evidence indicates several TRIM proteins serve as regulators in several signalling pathways in natural immunity. One of the TRIM proteins, TRIM56, has been reported to be a crucial regulator for viral infection and interferon signals.

In this paper, the authors showed that TRIM56 regulates the interferon signalling pathway. The authors found that TRIM56 interacts with cGAS and mono-ubiquitinates cGAS at the Lys335 and that TRIM56 exhibits dimerization of cGAS and DNA-binding activity and cGAMP production. These findings suggest that TRIM56 is a critical molecule for the cytosolic DNA sensing.

These insights are relevant in understanding the novel roles of TRIM56 in regulation of the innate immune system. This manuscript contains some important issues, such as the molecular insight of TRIM56 in the ubiquitination on cGAS. Although the experiments have been well performed, there may be some lacks of important controls and overstatements of results that preclude publication of the manuscript in the present form.

Specific Points

1. Fig. 3a: The results are not clear. The authors should quantitate the monoubiquitinated forms and analyze statistically using three individual experiments.

→ To examine the level of cGAS monoubiquitination with or without TRIM56 expression, the densities of the higher molecular-weight cGAS bands were quantitated in three independent experiments. This showed that cGAS monoubiquitination was increased ~6 fold upon TRIM56 expression. We have included these results to the revised manuscript.

2. Fig. 3b: To confirm that the band or smear is monoubiquitinated cGAS, the authors should show the molecular weight markers.

→ As the reviewer suggested, we added molecular weight markers to confirm the monoubiquitinated forms. These monoubiquitinated forms were shown near 75kDa. We have included these results to the revised manuscript.

3. Fig. 3d: We can see the slight poly (or multi)-ubiquitinated cGAS at lane 4. Can the authors rule out that polyubiquitinated cGAS by TRIM56 is important in DNA sensing system?

→ We repeated the ubiquitination assay with a stringent washing procedure (~500 mM NaCl) and observed monoubiquitinated forms. Furthermore, we have seen a single band shift, rather than formation of a ladder, in *in vitro* ubiquitination assays. This shows that TRIM56 preferentially monoubiquitinates cGAS. We have included these results to the revised manuscript.

4. In this study, the authors did not directly show that monoubiquitinated cGAS by TRIM56 enhances the dimerization and the binding activity to target DNA. It is very important to show the data substantiating this molecular insight.

→ As suggested, we conducted a SiMPull assay by adding the cGAS K335R mutant that no longer interacted with TRIM56. This showed that TRIM56 did not enhance the cGASK335R-DNA binding activity. This indicates that the cGAS dimerization induced by TRIM56 is important for its DNA binding activity. We have included these results to our revised manuscript.

Minor points:

1. The Method section (in vitro cGAS activity

assay and others): carefully check again. The authors put “space” between numeric values and unit.

→ We are sorry for our overlook. We have corrected it in the revised manuscript.

Reviewer #1 (Remarks to the Author):

The authors have satisfactorily answered the questions raised in the previous round. With the new modifications and data, the manuscript now offers a better understanding of the TRIM56 role on cGAS function. It is a very nice work.

Reviewer #2 (Remarks to the Author):

I have no more comments.